# ASTROFORMER: MORE DATA MIGHT NOT BE ALL YOU NEED FOR CLASSIFICATION

**Rishit Dagli**[1]
[1]University of Toronto
`rishit@cs.toronto.edu`

## ABSTRACT

Recent advancements in areas such as natural language processing and computer vision rely on intricate and massive models that have been trained using vast amounts of unlabelled or partly labeled data and training or deploying these state-of-the-art methods to resource constraint environments has been a challenge. Galaxy morphologies are crucial to understanding the processes by which galaxies form and evolve. Efficient methods to classify galaxy morphologies are required to extract physical information from modern-day astronomy surveys. In this paper, we introduce Astroformer, a method to learn from less amount of data. We propose using a hybrid transformer-convolutional architecture drawing much inspiration from the success of CoAtNet and MaxViT. Concretely, we use the transformer-convolutional hybrid with a new stack design for the network, a different way of creating a relative self-attention layer, and pair it with a careful selection of data augmentation and regularization techniques. Our approach sets a new state-of-the-art on predicting galaxy morphologies from images on the Galaxy10 DECals dataset, a science objective, which consists of 17736 labeled images achieving $94.86\%$ top-1 accuracy, beating the current state-of-the-art for this task by $4.62\%$. Furthermore, this approach also sets a new state-of-the-art on CIFAR-100 and Tiny ImageNet. We also find that models and training methods used for larger datasets would often not work very well in the low-data regime.

## 1 INTRODUCTION

Recently, many hybrid transformer-convolutional models have gained a lot of popularity in vision tasks, especially with the success of models like MobileViT (Mehta & Rastegari, 2022), ResNet-ViT (Dosovitskiy et al., 2021), ConViT (d'Ascoli et al., 2021), PatchConvNet (Touvron et al., 2021a), and CoAtNet (Dai et al., 2021). On one hand, the spatial inductive biases of convolutional neural networks allow them to learn representations with fewer parameters across different vision tasks as well as enable sample-efficient learning; however, they often have a potentially lower performance. On the other hand, transformers learn global representations and have shown a lot of success in vision tasks, and have outperformed convolutional neural nets for image classification. Though transformers tend to have larger model capacities, their generalization can be worse than convolutional neural networks. However, transformers are very data hungry (Hassani et al., 2021) and often require pre-training on large datasets. Typically Vision Transformers (ViTs) show great performance when trained on ImageNet-1k/22k (Deng et al., 2009) or JFT-300M (Sun et al., 2017) datasets; however, the absence of such large-scale pre-training is very detrimental to its performance as we show in this paper.

In practice, collecting high-quality labeled data or human annotators is very expensive. Synthetic data though shows significant promise but is often a distorted version of the real data and any modeling or inference performed on synthetic data comes with additional risks. Models trained on synthetic data often need to be fine-tuned with real data before being deployed (Tremblay et al., 2018; Jordon et al., 2022). Furthermore, methods such as transfer learning will not fully solve the problem often due to bias in pre-training datasets that do not reflect environments (Salman et al., 2023). We also explored semi-supervised learning approaches for this task using labeled data from the Galaxy10 DECals dataset (Leung & Bovy, 2019) and unlabelled galaxy images from the Galaxy-Zoo dataset (Lintott et al., 2011), this approach is promising and we believe is most certainly an

interesting direction. We however focused our efforts on building models when obtaining unlabeled data is costly as well Thus, in this paper, we focus on training in a supervised setting from scratch.

A galaxy's morphology describes its physical appearance and encodes important information about the physical processes that have shaped its growth (Parry et al., 2009). Galaxy morphologies are influenced heavily by the physical properties of galaxies, like star formation (Hashimoto et al., 1998; Sandage, 1986; Lotz et al., 2006) and galaxy mergers (Rodriguez-Gomez et al., 2017) among others (Buta, 2011). Thus understanding galaxy morphologies leads to a better understanding on these fronts as well. One such instance is where scientists have used the disc and spheroid structures that make up a galaxy to figure out crucial components of its formation (Deelman et al., 2004). As such, a galaxy's morphology is a culmination of internal physical processes (e.g., star formation, stellar dynamics such as active galactic nuclei and dark matter, feedback), as well as its environment and interactions with other galaxies. Thus, to fully comprehend the formation and evolution of galaxies, it is essential to accurately classify galaxy morphologies.

Our contributions can be summarized as: **(1)** We propose a hybrid transformer-convolutional architecture comparable to the approach employed by CoAtNet (Dai et al., 2021) with a different stack design and pair it with a careful selection of augmentation and regularization techniques which is able to learn and generalize well [1] in the low-data regime. **(2)** With our approach we establish a new state-of-the-art for the task of identifying galaxy morphologies from images on the Galaxy10 DECals dataset, and to the best of our knowledge, we believe this is the first work to use a hybrid transformer-convolutional model to improve models in this domain. We also set the new state-of-the-art for Tiny ImageNet and CIFAR-100. **(3)** We explore approaches and techniques for designing non-transfer learned models for the low-data regime in general which can be applied to tasks other than the one we explore.

## 2 DATA

We use the Galaxy10 DECals dataset introduced by Leung & Bovy (2019) which contains $\sim 17.7$k labeled images (see raw images in Appendix A.6, Figure 7). These images come from the DESI Legacy Imaging Surveys which include: two DECals campaign (Dey et al., 2019; Walmsley et al., 2022), the Beijing-Arizona Sky Survey (Zou et al., 2019) as well as the Mayall z-band Legacy Survey (Dey et al., 2019). The labels for the galaxy morphologies come from the GalaxyZoo Data Release 2 (Lintott et al., 2008; 2011). This dataset includes 10 strictly exclusive morphology classes: disturbed galaxies, merging galaxies, round smooth galaxies, in-between round smooth galaxies, cigar shaped smooth galaxies, barred spiral galaxies, unbarred tight spiral galaxies, unbarred loose spiral galaxies, edge-on galaxies without bulge, edge-on galaxies with bulge. The dataset is imbalanced, in that not all classes have a similar number of images. In particular, there are only 334 labeled images present for "cigar shaped smooth galaxies", so during training, we use stratified sampling. The size of this dataset also allows us to explore training techniques and develop approaches that work well in the low-data regime and generalize well.

The data collected for the DECals campaign uses Dark Energy Camera Flaugher et al. (2015) on the Blanco 4m telescope and covers both the North Galactic Cap region at Dec $\leq 32°$ and the South Galactic Cap region at Dec $\leq 34°$. The DECals survey utilizes a method of tiling the sky that involves three separate passes. These passes are slightly shifted in relation to one another, with an approximate offset range of $0.1°$ - $0.6°$. The specific pass and duration of exposure for each observation are determined in real-time based on multiple factors allowing for a nearly uniform level of depth across the survey. The Beijing-Arizona Sky Survey covers 5000 square degrees of the Northern Galactic Cap, using Steward Observatory's 2.3m Bok Telescope. The Mayall z-band Legacy Survey imaged the Dec $\geq 32°$ region using the 4-m Mayall telescope (Dey et al., 2019).

we also perform our experiments on the Tiny ImageNet (Le & Yang, 2015), CIFAR-100 (Krizhevsky et al., 2009),a dn CIFAR-10 (Krizhevsky et al., 2009). These datasets are very popularly used benchmarks for image classification in the low-data regime. Tiny ImageNet contains 100000 images of 200 classes (500 for each class) downsized to $64 \times 64$ colored images which are a subset of the ImageNet dataset (Deng et al., 2009). Each class has 500 training images, 50 validation images, and

---

[1]By generalization, we mean that we measured the gap between the training loss and the evaluation accuracy that describes how well the model can generalize to unseen data.

50 test images. The CIFAR-10 dataset consists of 60000 $32 \times 32$ color images in 10 classes, with 6000 images per class. The CIFAR-100 dataset is just like the CIFAR-10, except it has 100 classes containing 600 images each.

## 3 RELATED WORK

Classifying galaxy morphologies on the Galxy10 DECals is a rather well-established task and there have been multiple works in the past by Venn et al. (2019); Blancato (2020); Chen (2021); Ghadekar et al. (2022); Hui et al. (2022); Radhamani et al. (2022); HOLANDA & SANTOS (2022); Ćiprijanović et al. (2022); Maile et al. (2022) have employed multiple techniques for this task. However, no work has explored using hybrid models for this task which allowed us to set a new state-of-the-art for Galaxy10 DECals.

Transformer-convolutional hybrids are a recent innovation and in the past multiple models have proposed different approaches to constructing such hybrids namely: MobileViT (Mehta & Rastegari, 2022), ResNet-ViT (Dosovitskiy et al., 2021), ConViT (d'Ascoli et al., 2021), PatchConvNet (Touvron et al., 2021a), and CoAtNet (Dai et al., 2021). However, they do not explore the performance or modification that could be made to these methods to perform well for lower amounts of data with training from scratch. The work by Gani et al. (2022) extensively explores training ViTs in the low-data regime and have shown success on small datasets. Their work based on learning self-supervised inductive biases from small-scale datasets use these biases as a weight initialization scheme for fine-tuning. The work by Lee et al. (2021) explores modifying ViTs to learn locality inductive bias. In this paper, we explore training in low-data regimes with hybrid models and have motivated the use of hybrid models in the low-data regime, unlike these works.

## 4 METHODOLOGY

We develop a variant of the CoAtNet (Dai et al., 2021) model using a different stack design and careful selection of augmentation and regularization techniques. The Transformer block makes use of relative attention which efficiently combines depthwise convolutions (Sandler et al., 2018) and self-attention (Vaswani et al., 2017). A depthwise convolution uses a fixed kernel to extract features from a local region of the input data whereas self-attention allows the receptive field to be the global spatial space. Relative attention allows us to combine convolutions and self-attention.

$$y_i = \sum_{j \in \mathcal{G}} \frac{\exp\left(x_i^\top x_j + w_{i-j}\right)}{\sum_{k \in \mathcal{G}} \exp\left(x_i^\top x_k + w_{i-k}\right)} x_j \tag{1}$$

where $x_i, y_i \in \mathbb{R}^d$ are the input and output at position $i$, $w_{i-j}$ represents the depthwise convolution kernel and $\mathcal{G}$ represents the global spatial space. Here, the attention weight $A_{i,j}$ is decided by both $w_{i-j}$ and $x_i^\top x_j$. The update made to the attention weight is rather intuitive by simply summing a global static convolution kernel

$$\begin{aligned} A_{i,j} &= \sum_{k \in \mathcal{G}} \exp\left(x_i^\top x_k\right) \qquad \text{(standard self-attention)} \\ A_{i,j} &= \sum_{k \in \mathcal{G}} \exp\left(x_i^\top x_k + \boldsymbol{w_{i-k}}\right) \qquad \text{(relative self-attention)} \end{aligned} \tag{2}$$

To construct a network that uses relative attention, we adopt an approach similar to CoAtNets by first down-sampling the feature map via a multi-stage network with gradual pooling to reduce the spatial size and then employing the global relative attention. To do so, CoAtNets propose using a network of 5 stages (`S0`, `S1`, `S2`, `S3`, `S4`) where `S0` is a simple 2-layer convolutional Stem and `S1` employs Inverted Residual blocks (Sandler et al., 2018) with squeeze-excitation. In the work by Dai et al. (2021) they eliminate the possibility of using a C-C-C-T stack i.e. `S1`, `S2`, and `S3` employ Inverted Residual blocks (Sandler et al., 2018) and `S4` employs a Transformer block, due to supposedly low model performance. However, in our experiments, we find that a C-C-C-T design

Figure 1: Overview of the proposed model, notice the stack-design design we employ.

works much better than C-C-T-T which was adapted as the layout for CoAtNet. This is precisely due to **(1)** higher generalization capability of C-C-C-T and **(2)** highly unstable training of C-C-T-T and C-T-T-T architectures. These proposed architectural choices are summarized in Figure 1.

To evaluate the benefits of partially-occluded augmentation methods like CutMix (Yun et al., 2019), DropBlock (Ghiasi et al., 2018), and Cutout (DeVries & Taylor, 2017) hold for this task we run some experiments. We find that these regional dropout-based augmentation techniques have a strong detrimental effect on datasets such as the one we use [2] like the Galaxy10 DECals, this is mainly due to the nature of the task, one example we observe is that even some minor augmentations on an image with the class "edge-on galaxies with bulge" could cause the ground-truth (as identified by a human) of the augmented image to shift to the class "edge-on galaxies without bulge" [3] which is detrimental to the model performance. This example is visually in Appendix A.6, Figure 6.

Finally, for augmentation, we consider a combination of Mixup (Zhang et al., 2017) and RandAugment (Cubuk et al., 2020). As for regularization strategies, we make use of stochastic depth regularization, weight decay, and label smoothing. The hyperparameters values for these regularizations are listed in Appendix A.4. Surprisingly, we find that strong augmentations techniques give much higher performance gains than stronger regularization. Overall we believe, that judiciously choosing augmentation and regularization strategies is crucial to model performance in low-data regimes and careful selection of the associated hyperparameters for augmentation and regularization is equally important.

In brief, the reasons these models perform so well on low-data regime tasks, even when trained from scratch, and we believe it would work for other tasks in the low-data regime are: **(1)** Careful selection of augmentation and regularization is very important, especially in smaller datasets. **(2)** Our approach to train a hybrid transformer-convolutional model shows great generalizability and does not face the problem of highly unstable training and **(3)** The inherent translational equivalence helps make the training less prone to overfitting in the low-data regime, more experiments on this front were done by Mohamed et al. (2020). We postpone the proofs for this to Appendix A.2.

## 5 RESULTS

We report all results and compare them against previous models and a random baseline (equivalent to making a guess) in Table 1. We also provide a few random ground truth comparisons to our predictions in Figure 2. The performance of models is calculated using the metrics that are typical for an image classification problem, top-1 accuracy. Additionally, we find that using any of the other larger variants of CoAtNet is detrimental for this task and the model starts heavily overfitting even with our design changes, augmentation, and regularization. We also explore semi-supervised learning for this task, our experiments with semi-supervised learning techniques included applying Noisy Student (Xie et al., 2020), SimCLRv2 (Chen et al., 2020), and Meta Pseudo Labels (Pham et al., 2021) to the Galaxy10 DECals dataset as well as using unlabeled images from the GalaxyZoo Dataset though. These results are also summarized in Table 1.

---

[2]We also explore Attentive CutMix (Walawalkar et al., 2020) and though it identifies the most discriminative regions based on the intermediate attention maps from a feature extractor, we do not observe a significantly higher generalization capability.

[3]More information on how galaxy morphologies are manually classified can be found at `https://data.galaxyzoo.org/gz_trees/gz_trees.html`

Table 1: Model Performance on Galaxy10 DECals dataset. `Galaxy10 only` denotes training on the Galaxy10 DECals dataset only; `Galaxy10 + Zoo` denotes the use of extra unlabeled data from GalaxyZoo, though the models that use extra unlabeled data are not the focus of this work, we present those results as well. The rows under the horizontal line in this table represent experiments performed in this work.

| Method Description | Galaxy10 DECals top-1 accuracy($\uparrow$) | |
| --- | --- | --- |
| | Galaxy10 only | Galaxy10 + Zoo |
| Random Baseline | 14.77 | - |
| Fractal Analysis (Radhamani et al., 2022) | 73.45 | - |
| Architectural Optimization Over Subgroups (Maile et al., 2022) | 77.00 | - |
| DeepAstroUDA (Ćiprijanović et al., 2022) | 79.00 | - |
| Deep Galaxies CNN (Ghadekar et al., 2022) | 84.04 | - |
| EfficientNet (Tan & Le, 2019) | 86.00 | - |
| Luma (HOLANDA & SANTOS, 2022) | 86.20 | - |
| DenseNet 121 (Iandola et al., 2014) | 88.64 | - |
| EfficientNetv2 (Tan & Le, 2021) | 90.24 | - |
| Noisy Student (Xie et al., 2020) | - | **91.80** |
| SimCLRv2 (Chen et al., 2020) | - | 90.24 |
| Standard CoAtNet-4 (Dai et al., 2021) | 81.55 | - |
| **Ours (Astroformer)** | **94.86** | - |

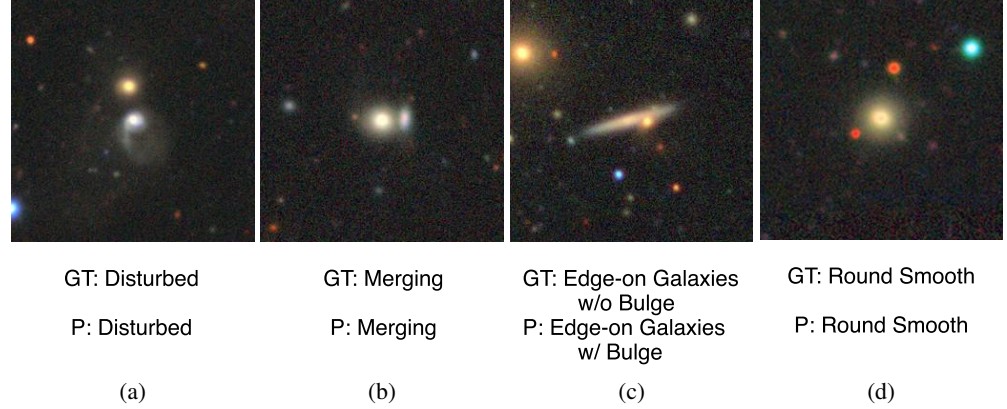

| GT: Disturbed | GT: Merging | GT: Edge-on Galaxies w/o Bulge | GT: Round Smooth |
| P: Disturbed | P: Merging | P: Edge-on Galaxies w/ Bulge | P: Round Smooth |
| (a) | (b) | (c) | (d) |

Figure 2: Random selection of ground truth comparison against our model predictions on the hold-out test set. We also include one example where the model predicted incorrectly (c). The acronyms "GT" and "P" in this figure refer to the ground truth and predicted labels.

## 6    DISCUSSION

In this paper, we developed a supervised model to train from scratch on the Galaxy10 DECals dataset and explored other methods to train from scratch in the low-data regime. We train a CoAtNet model and pair it with a careful selection of augmentation and regularization strategies as well as use a different stack design which was earlier thought to have severely low model performance and a novel way of creating relative self-attention layers on top of the CoAtNet model (Dai et al., 2021). We apply this proposed model in the low-data regime and achieve state-of-the-art performance on the Galaxy10 DECals dataset. Furthermore, with this approach, we also establish new state-of-the-art without using extra training data on popular low-data regime image classification datasets, CIFAR-100 (Krizhevsky et al., 2009) and Tiny ImageNet (Le & Yang, 2015), and competitive results on CIFAR-10 (Krizhevsky et al., 2009) as indicated in Appendix A.3. From the perspective of science objectives, we believe this model will enable more precise studies of galaxy morphology. In the future, we hope to see more methods for training models with low data from scratch and we believe our model is a potential choice for a variety of other low-data regime tasks. We also hope that this model could potentially be used as a backbone for other vision tasks in the low-data regime.

## ACKNOWLEDGEMENTS

The authors would like to thank Google for supporting this work by providing Google Cloud credits. The authors would also like to thank Google TPU Research Cloud (TRC) program [4] for providing access to TPUs.

We thank David Lindell of the University of Toronto for insightful conversations. We thank Caleb Lammers, Jo Bovy, and Henry Leung of the University of Toronto for insightful discussions and their domain expertise in the area of galaxy morphologies and evolution.

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

# A APPENDIX

## A.1 EXTENDED RELATED WORK

**Self-Attention for Vision.** Self Attention has extensively been applied to various computer vision tasks, such as image classification, object detection, and semantic segmentation. The idea of self-attention was first introduced by Vaswani et al. (2017), where they proposed the Transformer model for neural machine translation. The Transformer model consists of an encoder and a decoder, each composed of multiple layers of self-attention and feed-forward networks. The self-attention mechanism can be seen as a generalization of the attention mechanism that was previously used in conjunction with RNNs for sequence modeling. The application of self-attention and transformers to computer vision tasks faces some challenges due to the high-resolution and spatial structure of visual data. One challenge is the quadratic computational complexity of self-attention, which limits its scalability to large inputs. Another challenge is the lack of local information in self-attention, which may hinder its ability to capture fine-grained details and local patterns in images. To address these challenges, several variants and extensions of self-attention and transformers have been proposed for computer vision tasks. For example, Dosovitskiy et al. (2021) proposed Vision Transformer (ViT), which applies a vanilla Transformer to image classification by dividing an image into patches and treating them as tokens.

These are some examples of how self-attention and transformers can be adapted and improved for computer vision tasks. For a more comprehensive review of vision transformers, we refer readers to the dedicated surveys by Khan et al. (2022) and Han et al. (2023).

**Hybrid Models.** Hybrid convolution-attention models are a recent trend in computer vision that aim to combine the advantages of CNNs and self-attention mechanisms. CNNs are known for their ability to capture local features and spatial invariance, while self-attention can model long-range dependencies and global context. However, CNNs and self-attention have different strengths and weaknesses, and finding the optimal balance between them is not trivial.

One approach to hybridize convolution and attention is to augment the CNN backbone with explicit self-attention or non-local modules (Wang et al., 2018), or to replace certain convolution layers with standard self-attention (Bello et al., 2019) or a more flexible mix of linear attention and convolution. These methods often improve the accuracy of CNNs, but they also increase the computational cost and complexity. Moreover, they do not exploit the natural connection between depthwise convolution and relative attention, which can be fused together to form a more efficient and effective hybrid layer (Dai et al., 2021). Another approach to hybridize convolution and attention is to start with a Transformer backbone and try to incorporate explicit convolution or some desirable properties of convolution into the Transformer layers (Liu et al., 2021b). These methods aim to overcome the limitations of vanilla Transformers, such as the lack of inductive biases, the quadratic complexity, and the dependence on large-scale pre-training. However, they also face challenges such as how to design convolutional embeddings, how to integrate convolutional operations into the attention mechanism, and how to balance the trade-off between model capacity and generalization.

A recent work that proposes a novel hybrid convolution-attention model is CoAtNet (Dai et al., 2021), which is based on two key insights: (1) by using simple relative attention, depthwise convolution, and self-attention can be naturally fused together to form a hybrid layer called CoAt layer; (2) by stacking CoAt layers and standard convolution layers in a principled manner, generalization, capacity, and efficiency can be dramatically improved.

## A.2 MODEL DETAILS

**Inherent Translation Equivariance.**

**Theorem 1.** *A variant of relative attention:*

$$y_i = \sum_{j \in \mathcal{G}} \frac{\exp\left(x_i^\top x_j + w_{i-j}\right)}{\sum_{k \in \mathcal{G}} \exp\left(x_i^\top x_k + w_{i-k}\right)} x_j \tag{3}$$

*where $x_i, y_i \in \mathbb{R}^d$ are the input and output at position $i$, $w_{i-j}$ represents the depthwise convolution kernel and $\mathcal{G}$ represents the global spatial space preserves translational equivariance.*

*Proof.* Informally we can understand this proof as identifying that the depthwise convolution kernel for any position pair $(i, j)$ is only dependent on $i - j$, the relative positions rather than the values of $i$ and $j$ individually.

We will also use the well-known result that convolutions enjoy translational equivariance, this also aligns well with the general nature of tasks in vision and also partly helps generalize the model to different positions or to images of different sizes.

Formally we can define $T_k$, to be the translation operator that shifts the input sequence $x$ by $k$ positions, that is, $T_k(x)_i = x_{i-k}$. Similarly, let $T_k(y)_i$ denote the output of the relative attention mechanism when the input is shifted by $k$ positions. We then want to show that $T_k(y)_i = y_{i-k}$ for all $i$. That is a shift in the input sequence results in an equivalent shift in the output sequence. Equivalently for this proof, we use the simpler notation where we let $x'_i = x_i + \Delta$ and $x'_j = x_j + \Delta$ be the translated input vectors. We want to show that $y'_i = y_i + \Delta$.

Substituting the translated input vectors into Equation 3, we have:

$$y'_i = \sum_{j \in \mathcal{G}} \frac{\exp\left((x_i + \Delta)^\top (x_j + \Delta) + w_{i-j}\right)}{\sum_{k \in \mathcal{G}} \exp\left((x_i + \Delta)^\top x_k + w_{i-k}\right)} x_j \tag{4}$$

Expanding the dot product, we get:

$$\begin{aligned}
(x_i + \Delta)^\top (x_j + \Delta) &= x_i^\top x_j + x_i^\top \Delta + \Delta^\top x_j + \Delta^\top \Delta \\
&= x_i^\top x_j + 2\Delta^\top x_j + \Delta^\top \Delta \\
&= (x_i^\top x_j + w_{i-j}) + 2\Delta^\top x_j + \Delta^\top \Delta
\end{aligned} \tag{5}$$

Substituting this back into Equation 4, we get:

$$\begin{aligned}
y'_i &= \sum_{j \in \mathcal{G}} \frac{\exp\left((x_i^\top x_j + w_{i-j}) + 2\Delta^\top x_j + \Delta^\top \Delta\right)}{\sum_{k \in \mathcal{G}} \exp\left((x_i + \Delta)^\top x_k + w_{i-k}\right)} x_j \\
&= \sum_{j \in \mathcal{G}} \frac{\exp\left(x_i^\top x_j + w_{i-j}\right) \exp\left(2\Delta^\top x_j + \Delta^\top \Delta\right)}{\sum_{k \in \mathcal{G}} \exp\left(x_i^\top x_k + w_{i-k}\right) \exp\left(\Delta^\top x_k\right)} x_j \\
&= \sum_{j \in \mathcal{G}} \frac{\exp\left(x_i^\top x_j + w_{i-j}\right)}{\sum_{k \in \mathcal{G}} \exp\left(x_i^\top x_k + w_{i-k}\right)} \exp\left(\Delta^\top x_j\right) \left(\frac{\exp\left(\Delta^\top \Delta\right)}{\sum_{k \in \mathcal{G}} \exp\left(\Delta^\top x_k\right)} x_j\right) \\
&= y_i + \Delta \sum_{j \in \mathcal{G}} \frac{\exp\left(x_i^\top x_j + w_{i-j}\right)}{\sum_{k \in \mathcal{G}} \exp\left(x_i^\top x_k + w_{i-k}\right)} \exp\left(\Delta^\top x_j\right)
\end{aligned} \tag{6}$$

Thus, we can write this as:

$$y'_i = y_i + \Delta \sum_{j \in \mathcal{G}} \frac{\exp\left(x_i^\top x_j + w_{i-j}\right)}{\sum_{k \in \mathcal{G}} \exp\left(x_i^\top x_k + w_{i-k}\right)} \exp\left(\Delta^\top x_j\right) \tag{7}$$

Notice that the term inside the sum on the right-hand side of Equation 7 is equivalent to the attention weight between $x'_i$ and $x'_j$. Therefore, we can rewrite the sum as:

$$\sum_{j \in \mathcal{G}} \frac{\exp\left(x_i^\top x_j + w_{i-j}\right)}{\sum_{k \in \mathcal{G}} \exp\left(x_i^\top x_k + w_{i-k}\right)} \exp\left(\Delta^\top x_j\right) = \sum_{j \in \mathcal{G}} \frac{\exp\left(x_i'^\top x'_j + w_{i'-j'}\right)}{\sum_{k \in \mathcal{G}} \exp\left(x_i'^\top x'_k + w_{i'-k'}\right)} \tag{8}$$

where $i' - j'$ and $i' - k'$ are the relative positions between $x'_i$ and $x'_j$ and $x'_k$, respectively.

Therefore, we can rewrite Equation 7 as:

$$y_i' = \sum_{j \in \mathcal{G}} \frac{\exp\left(x_i'^{\top} x_j' + w_{i'-j'}\right)}{\sum_{k \in \mathcal{G}} \exp\left(x_i'^{\top} x_k' + w_{i'-k'}\right)} x_j' = y_i + \Delta \tag{9}$$

which shows that the output $y_i$ is translated by the same vector $\Delta$ as the input $x_i$ and $x_j$, and therefore the relative attention in Equation 3 enjoys the property of translational equivariance. $\qquad \square$

**Theorem 2.** *The variant of relative attention described in Equation 3 has the ability to do input-adaptive weighting.*

*Proof.* To show this, in this proof, we will show that the form in Equation 3 that the attention weight $A_{i,j}$ is influenced by the input $x_i$ in a way that adapts to the input.

We can rewrite the numerator of the attention weight as:

$$\exp\left(x_i^{\top} x_j + w_{i-j}\right) = \exp\left(x_i^{\top} x_j\right) \exp\left(w_{i-j}\right) \tag{10}$$

Since the depthwise convolution kernel $w_{i-j}$ is fixed for all inputs, it does not adapt to the input. Therefore, the input-adaptive property of the attention weight must come from the term $\exp\left(x_i^{\top} x_j\right)$.

Now, we can write the denominator of the attention weight as:

$$\sum_{k \in \mathcal{G}} \exp\left(x_i^{\top} x_k + w_{i-k}\right) = \sum_{k \in \mathcal{G}} \exp\left(x_i^{\top} x_k\right) + \exp\left(w_{i-k}\right) \tag{11}$$

Again, the term $\exp\left(w_{i-k}\right)$ is fixed and does not adapt to the input. Therefore, we only need to focus on the term $\exp\left(x_i^{\top} x_k\right)$ to see if it adapts to the input.

$$A_{i,j} = \frac{\exp w_{i-j}}{\sum_{k \in \mathcal{G}} \exp\left(x_i^{\top} x_k - x_i^{\top} x_j + w_{i-k}\right)} \tag{12}$$

Now, we can see that the term $\exp\left(x_i^{\top} x_k - x_i^{\top} x_j\right)$ represents the similarity between the input vectors $x_i$ and $x_k$ relative to the similarity between $x_i$ and $x_j$. This relative similarity term ensures that the attention weight adapts to the input, as it depends on the relationship between the input vectors rather than their absolute values. $\qquad \square$

**Pre-Activation.** We follow the same pre-activation structure as demonstrated by Dai et al. (2021) for both the Inverted Residual blocks and Transformer blocks:

$$x \leftarrow x + \texttt{Module}(\texttt{Norm}(x)) \tag{13}$$

where `Module` denotes the Inverted Residual, Self-Attention, or FFN module, while `Norm` corresponds to `BatchNorm` for Inverted Residual block and `LayerNorm` for Self-Attention and FFN.

**Down-Sampling.** We follow the same down-sampling structure as demonstrated by Dai et al. (2021). For the first block inside each stage from S1 to S4, down-sampling is performed independently for the residual branch and the identity branch. The down-sampling self-attention module can is expressed as:

$$x \leftarrow \texttt{Proj}(\texttt{Pool}(x)) + \texttt{Attention}(\texttt{Pool}(\texttt{Norm}(x))) \tag{14}$$

As for the Inverted Residual block, the down-sampling in the residual branch is instead achieved by using a stride-2 convolution to the normalized inputs

$$x \leftarrow \texttt{Proj}(\texttt{Pool}(x)) + \texttt{Conv}(\texttt{DepthConv}(\texttt{Conv}(\texttt{Norm}(x), \texttt{ stride = 2}))) \tag{15}$$

## A.3 MODEL PERFORMANCE ON OTHER LOW-DATA REGIME DATASETS

In this section, we explore applying the proposed model to other low-data regime image classification tasks namely CIFAR-100 (Krizhevsky et al., 2009), Tiny ImageNet (Le & Yang, 2015), and CIFAR-10 (Krizhevsky et al., 2009) without extra training data. Similar to the experiments for Galaxy10 DECals, the hyperparameters for these tasks were also found through a naive hyperparameter search, however, any of the models trained on these datasets do not use stratified sampling. In Table 2 we report our results for CIFAR-100, in Table 3 for Tiny ImageNet, and in Table 4 for CIFAR-10 datasets.

Table 2: Model Performance on the CIFAR-100 dataset. We only present models that do not use extra unlabeled data which can be compared with these experiments. We notice that even without using extra training data our approach beats multiple well-established models which use extra-training data.

| Method Description | CIFAR-100 top-1 accuracy(↑) | Extra Training Data |
|---|---|---|
| EfficientNetV2 (Tan & Le, 2021) | 92.30 | ✓ |
| TResNet (Ridnik et al., 2021) | 93.00 | ✓ |
| CaiT (Touvron et al., 2021b) | 93.10 | ✓ |
| μ2Net (Gesmundo & Dean, 2022) | 94.95 | ✓ |
| ML-Decoder (Ridnik et al., 2023) | 95.10 | ✓ |
| EffNet-L2 (SAM) (Chen et al., 2021) | **96.08** | ✓ |
| CoAtNet-5 (Dai et al., 2021) | 81.21 | ✗ |
| CoAtNet-4 (Dai et al., 2021) | 84.60 | ✗ |
| WRN (Zhao et al., 2022a) | 86.90 | ✗ |
| DenseNet (Iandola et al., 2014) | 87.44 | ✗ |
| ColorNet (Gowda & Yuan, 2019) | 88.40 | ✗ |
| ShakeDrop (Cubuk et al., 2018) | 89.30 | ✗ |
| PyramidNet (Zhao et al., 2022b) | 89.90 | ✗ |
| **Ours (Astroformer)** | **93.36** | ✗ |

Table 3: Model Performance on the Tiny ImageNet dataset. We only present models that do not use extra unlabeled data which can be compared with these experiments. We notice that even without using extra training data our approach beats multiple well-established models which use extra-training data.

| Method Description | TI top-1 accuracy(↑) | Extra Training Data |
|---|---|---|
| EfficientNet (DCL) (Luo et al., 2019) | 84.39 | ✓ |
| ViT (PUGD) (Tseng et al., 2022) | 90.74 | ✓ |
| DeiT (PUGD) (Tseng et al., 2022) | 91.02 | ✓ |
| Swin-L (Huynh, 2022) | **91.35** | ✓ |
| PreActResNet (Ramé et al., 2021) | 70.24 | ✗ |
| ResNeXt-50 (SAMix+DM) (Liu et al., 2022) | 72.39 | ✗ |
| Context-Aware Pipeline (Yao et al., 2021) | 73.60 | ✗ |
| WaveMixLite (Jeevan et al., 2023) | 77.47 | ✗ |
| DeiT (Lutati & Wolf, 2022) | 92.00 | ✗ |
| **Ours (Astroformer)** | **92.98** | ✗ |

## A.4 IMPLEMENTATION DETAILS

In this section, we explain the implementation details of the experiments and our proposed model.

**Sampling.** There is a class imbalance in the dataset, meaning that not all classes have a similar number of images. For this reason, we follow a stratified sampling strategy during data loading to ensure each batch contains $10 \pm 4\%$ instances of each label class.

Table 4: Model Performance on the CIFAR-10 dataset. We only present models that do not use extra unlabeled data which can be compared with these experiments. We notice that even without using extra training data our approach beats multiple well-established models which use extra-training data.

| Method Description | CIFAR-10 top-1 accuracy(↑) | Extra Training Data |
|---|---|---|
| CeiT (Yuan et al., 2021) | 99.10 | ✓ |
| ViT (PUGD) (Tseng et al., 2022) | 99.13 | ✓ |
| BIT-L (Kolesnikov et al., 2020) | 99.37 | ✓ |
| CaiT (Touvron et al., 2021b) | 98.40 | ✓ |
| CvT (Wu et al., 2021) | **98.39** | ✓ |
| ViT-SAM (Chen et al., 2021) | 98.60 | ✗ |
| PyramidNet (Zhao et al., 2022b) | 98.71 | ✗ |
| LaNet (Wang et al., 2021) | 99.03 | ✗ |
| **Ours (Astroformer)** | 99.12 | ✗ |
| μ2Net (Gesmundo & Dean, 2022) | 99.49 | ✗ |
| ViT (Dosovitskiy et al., 2021) | **99.50** | ✗ |

**Model backbone.** Throughout this work we use CoAtNet as the model backbone due to their success in efficiently unifying depthwise convolutions and self-attention through relative attention and their approach of vertically stacking convolution layers and attention layers. Since the Galaxy10 DECals dataset does not contain a large amount of data, other transformer-based models did not show great results for this task whereas a hybrid model with regularization and augmentation techniques, was able to generalize well and achieved better results.

**Baseline model.** We established a naive baseline with random guesses. The baseline model we chose was training CoAtNet-4 without any design modifications. This gets to a top-1 accuracy of 81.55% on the Galaxy10 DECals dataset. To train this model, we employ standard augmentations (MixUp and CutMix) and train the network for 300 epochs using the default settings in `timm` with a batch size of 256.

**Loss function.** We adopt the standard cross entropy loss with smoothing. We also performed some preliminary experiments using Dense Relative Localization Loss (Liu et al., 2021a) and we believe this might be a potentially promising direction as well.

**Code.** Our code is in PyTorch 1.10 (Paszke et al., 2019). We use a number of open-source packages to develop our training workflows. Most of our experiments and models were trained with `timm` (Wightman, 2019) and we also use `mmclassify` (Contributors, 2020) for some of the experiments. Our hardware setup for the experiments included either four NVIDIA Tesla V100 GPUs or a TPUv3-8 cluster. We utilized mixed-precision training with PyTorch's native AMP (through `torch.cuda.amp`) for mixed-precision training and a distributed training setup (through `torch.distributed.launch`) which allowed us to obtain significant boosts in the overall model training time. We report the number of parameters and FLOPS of the final model in Table 6.

**Hyperparameters.** The choice of hyperparameters for training the Astroformer model is shown in Table 5. The rest of the hyperparameters were kept to their defaults as provided in `timm` (Wightman, 2019). The hyperparameters related to Lookahead (Zhang et al., 2019) are used at their default values as suggested.

A.5  FULL TABULAR RESULTS

We provide Tables 6-9 that provides all the numerical results.

Table 5: Hyper-parameters used to train Astroformer for Galaxy10 DECals.

| Hyper-parameter | Values |
|---|---|
| Stochastic depth rate | 0.2 |
| Center crop | False |
| Mixup Alpha | 0.8 |
| Train epochs | 300 |
| Label smoothing | 0.1 (`timm` default) |
| Train batch size | 256 |
| Optimizer type | Lookahead (Zhang et al., 2019) + RAdam (Liu et al., 2020) |
| LR decay schedule | Cosine |
| Base learning rate | 2e-5 |
| Warmup learning rate | 1e-5 (`timm` default) |
| Warmup | 5 epochs |
| Weight decay rate | 1e-2 |
| Gradient clip | None |
| EMA decay rate | None |
| RandAugment layers | 2 |

Table 6: Sorted table based on the number of parameters in each family for top-1 accuracy and scaling curves of multiple models on the Galaxy10 DECals dataset shown in Figure 3.

| Model family | Params (M) | FLOPs (G) | Top-1 Accuracy (↑) |
|---|---|---|---|
| **Astroformer** | 655.04 | 115.97 | 75.27 |
| | 271.54 | 60.54 | 94.86 |
| | 161.75 | 31.36 | 92.39 |
| **SwinV2** | 274.06 | 81.55 | 81.55 |
| | 195.16 | 35.09 | 91.23 |
| | 86.86 | 15.86 | 84.58 |
| **ViT** | 1011.22 | 267.18 | 78.27 |
| | 630.78 | 167.40 | 78.46 |
| | 303.31 | 61.60 | 80.25 |
| | 87.46 | 4.41 | 84.36 |
| | 85.81 | 17.58 | 84.35 |
| | 5.53 | 1.26 | 74.21 |
| **EfficientNet v2** | 117.25 | 12.40 | 90.12 |
| | 52.87 | 5.46 | 85.67 |
| | 20.19 | 2.91 | 84.37 |
| **CoAtNet** | 163.08 | 36.69 | 82.39 |
| | 72.62 | 16.58 | 87.45 |
| | 40.87 | 8.76 | 85.24 |
| **Swin** | 195.01 | 34.53 | 93.21 |
| | 86.75 | 15.47 | 82.38 |
| | 23.68 | 4.63 | 81.57 |

Table 7: Contribution to the error rate for Astroformer-4 on the Galaxy10 Dataset according to the classes in the test set.

| Class | Contribution to error rate (%) |
|---|---|
| 0 | 0.31 |
| 1 | 0.23 |
| 2 | 1.28 |
| 3 | 0.3 |
| 4 | 1.02 |
| 5 | 0.14 |
| 6 | 0.31 |
| 7 | 0.12 |
| 8 | 1.16 |
| 9 | 0.27 |
| | 5.14 |

## A.6 SUPPLEMENTAL FIGURES

We provide Figures 3-2 that provides supplementary images.

Table 8: Sorted table based on the number of parameters in each family for top-1 accuracy and scaling curves of multiple models on the CIAFR-100 shown in Figure 4.

| Model family | Params (M) | FLOPs (G) | Top-1 Accuracy (↑) |
|---|---|---|---|
| **EfficientNetV2** | 117.36 | 12.4 | 92.3 |
| | 52.98 | 5.45 | 92.2 |
| | 20.3 | 2.91 | 91.5 |
| **ViT** | 305.61 | 15.38 | 74.26 |
| | 303.4 | 61.6 | 75.28 |
| | 87.53 | 4.41 | 86.3 |
| | 85.87 | 17.58 | 87.1 |
| **EfficientNet** | 84.87 | 7.21 | 92.32 |
| | 64.04 | 5.36 | 91.7 |
| | 40.96 | 3.5 | 89.96 |
| | 28.54 | 2.47 | 88.76 |
| | 17.72 | 1.58 | 88.72 |
| **CoAtNet** | 271.68 | 62.65 | 92.13 |
| | 163.22 | 36.68 | 89.45 |
| | 72.71 | 16.58 | 87.9 |
| **Astroformer** | 655.34 | 115.97 | 89.38 |
| | 271.68 | 60.54 | 93.36 |
| | 161.95 | 31.36 | 87.65 |

Table 9: Sorted table based on the number of parameters in each family for top-1 accuracy and scaling curves of multiple models on the CIAFR-10 shown in Figure 5.

| Model family | Params (M) | FLOPs (G) | Top-1 Accuracy (↑) |
|---|---|---|---|
| **EfficientNetV2** | 117.25 | 12.40 | 99.1 |
| | 52.87 | 5.46 | 99.0 |
| | 20.19 | 2.91 | 98.7 |
| **ViT** | 305.52 | 15.39 | 77.8 |
| | 303.31 | 61.60 | 76.5 |
| | 85.81 | 17.58 | 74.9 |
| | 87.46 | 4.41 | 73.4 |
| **EfficientNet** | 63.81 | 5.37 | 99.0 |
| | 40.76 | 3.51 | 97.23 |
| | 28.36 | 2.47 | 95.43 |
| | 17.57 | 1.59 | 93.89 |
| | 10.71 | 1.03 | 93.45 |
| **CoAtNet** | 72.62 | 16.58 | 92.17 |
| | 163.08 | 36.69 | 91.43 |
| | 271.54 | 62.65 | 91.34 |
| **Astroformer** | 161.75 | 31.36 | 99.12 |
| | 271.54 | 60.54 | 98.93 |
| | 655.04 | 115.97 | 93.23 |

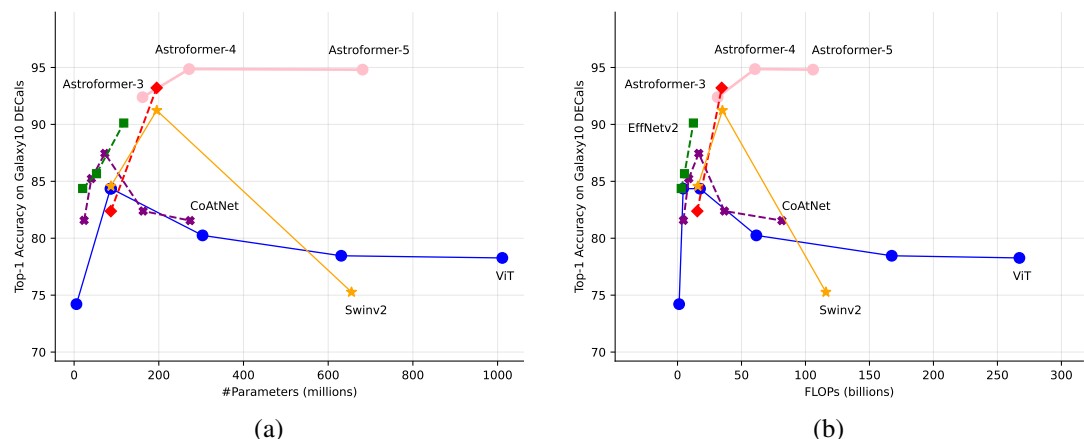

Figure 3: (a) The top-1 accuracy to parameter scaling curves for multiple models on the Galaxy10 DECals dataset. (b) The top-1 accuracy to FLOPs scaling curves for multiple models on the Galaxy10 DECals dataset. All these scaling curves are for the evaluation size of $224^2$. The data for this graph can be found in Table 6

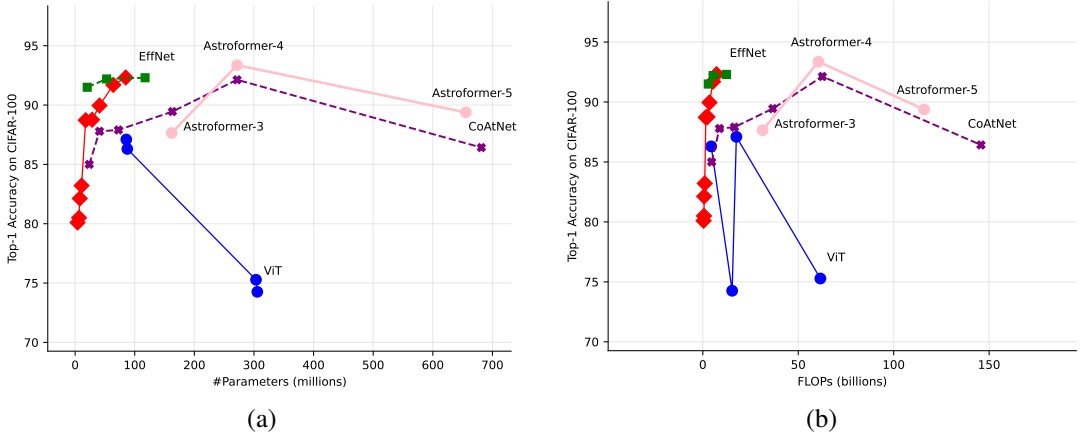

Figure 4: (a) The top-1 accuracy to parameter scaling curves for multiple models on the CIFAR-100 dataset. (b) The top-1 accuracy to FLOPs scaling curves for multiple models on the CIFAR-100 dataset. All these scaling curves are for the evaluation size of $224^2$. The data for this graph can be found in Table 8

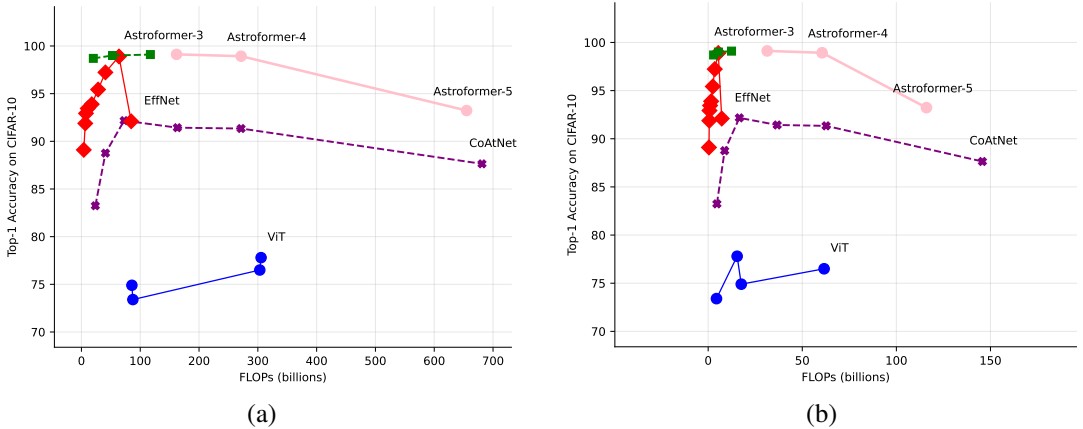

(a)                                    (b)

Figure 5: (a) The top-1 accuracy to parameter scaling curves for multiple models on the CIFAR-10 dataset. (b) The top-1 accuracy to FLOPs scaling curves for multiple models on the CIFAR-10 dataset. All these scaling curves are for the evaluation size of $224^2$. The data for this graph can be found in Table 9

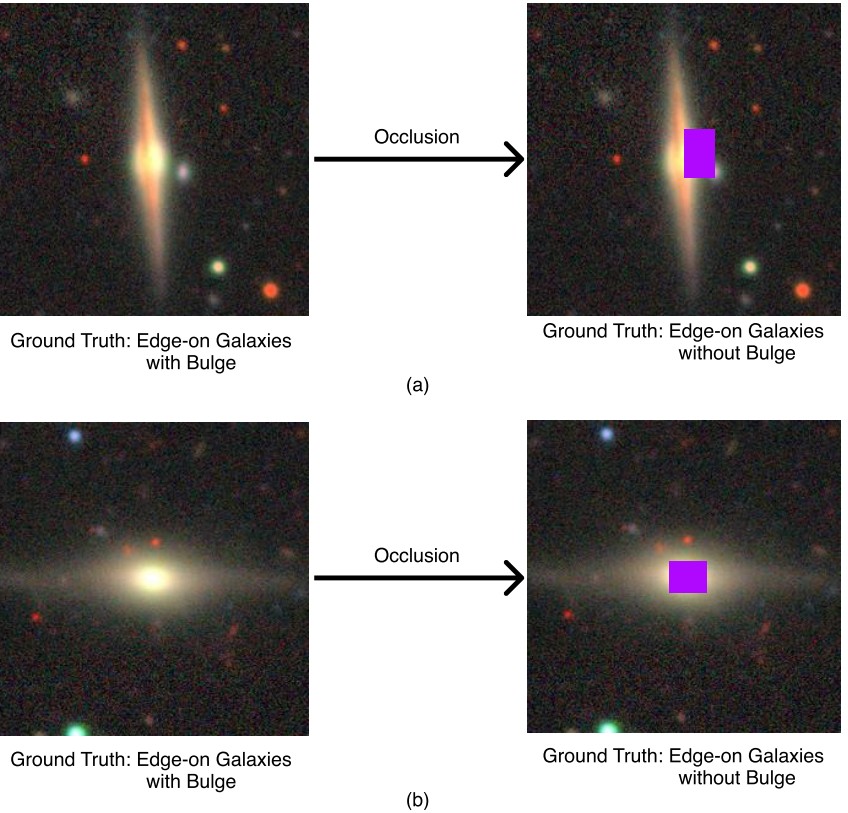

Figure 6: Random selection of augmented images after applying regional dropout-based augmentation techniques. We note the visible differences in the galaxy morphologies as well as observe visually why partial-occluded augmentation techniques do not work well for this dataset.

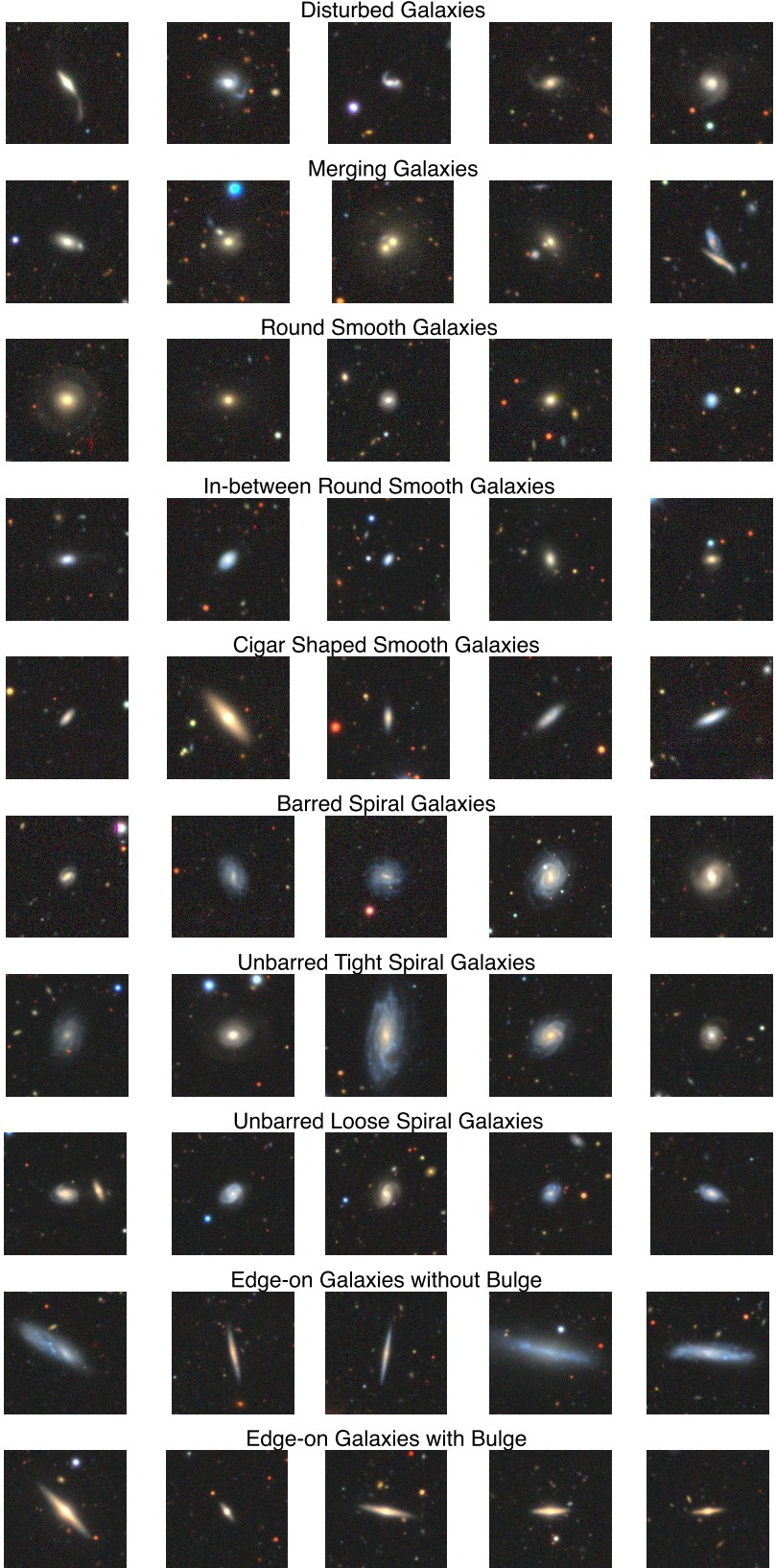

Figure 7: Raw data (galaxy images and labels) from the Galaxy10 DECals dataset. All images are a random sample from the training set.

