# OpenReview forum: "Astroformer: More Data might not be all you need for Classification"
_ICLR.cc/2023/TinyPapers — Submitted to Tiny Papers @ ICLR 2023_

### Official Review · Reviewer_LmDq · 2023-03-29

**Confidence:** 3

**Summary Of Contributions:**

The paper proposes to apply a transformer-convolutional network for the task of Galaxy Morphologies prediction. It reaches state-of-the-art performance in a low-data regime.

**Rating:**

Clear, Correct, and Reproducible (CCR): a submission which meets the reviewing criteria

**Strengths And Weaknesses:**

STRENGTHS
=======
1) The paper deploys a recent technique in a new context, obtaining promising results and pointing a direction for the community.
2) The introduction is clear, and the problem and the contribution well stated

WEAKNESSES
=========
1) The paper does an excellent job of introducing the reader, and the appendix complements a lot of further information. However, I think some more discussion of Galaxies Morphologies would be good to ease the reader's understanding of the challenge of the task from a technical perspective. Are the labels reported in Figure 4 all the possible morphologies? Are the galaxies always somehow centred in the image?
2) The title wording is interesting, but it might be more focused; I will leave a suggestion below
3) In discussion, future works mention that other low-regime tasks might benefit from this direction; but I think would be good to have some more insights designed for the galaxy morphology fields: what is the impact of having such a better prediction? What is the use case? (e.g., are astrophysics already using these tools, or are performance still too low, and this paper moves an essential step in this direction?)


Minor:
Typo: "... costly as well Thus, in this paper ..." -> period is missing

**Suggested Changes:**

To improve the paper, I suggest to:
1) Add more discussion about the technical difficulties of the considered domain, and highlight the applicative relevance of the proposed approach, both in the introduction and in the discussion sections
2) Include a more sharp and more direct title. In my opinion, the key elements of the paper are a) Transformer-Convolutional architecture, b) Galaxy Morphologies prediction, c) low data regime. So an option could be:  "AstroFormer: A Transformer-Convolutional method to predict Galaxy Morphologies in a low data regime".

---

### Official Review · Reviewer_hm8k · 2023-03-31

**Confidence:** 4

**Summary Of Contributions:**

The authors propose a hybrid transformer-convolutional architecture with a different stack design for the network to effectively predict Galaxy Morphologies from images. The author identifies that the success of this combination depends on the careful selection of data augmentation and regularization techniques. They further claimed to beat the current state-of-the-art model by 4.6% on the Galaxy10 DECal dataset.

**Rating:**

High Potential (HP): a submission which meets the reviewing criteria and has potential to make an impact on the field

**Strengths And Weaknesses:**

- Clarity: The paper was easy to follow through and relevant related works are cited.

- Correctness:
    - No results are presented in the main papers. How can the performance of the proposed model be judged?
    - The technique seems to be highly sensitive to augmentation and regularization techniques. Will the proposed model be useful in real-world settings?
    - The motivation for the work is clear.

- Reproducibility: No code is provided but I see enough details in Appendix to reproduce the analysis and the authors promise to release the code upon acceptance.

- Follows basic requirements: yes

**Suggested Changes:**

- Does this hybrid technique tested for models other than CoAtNet?
- What do you need to change in the current design to make it work for other models?

---

### Official Review · Reviewer_xjBS · 2023-04-02

**Confidence:** 4

**Summary Of Contributions:**

This paper proposes a hybrid transformer-convolutional architecture and data augmentation techniques to efficiently classify galaxy morphologies with limited data, setting a new state-of-the-art on the Galaxy10 DECals dataset.

**Rating:**

Great Start (GS): a submission which meets some of the reviewing criteria but has room for improvement

**Strengths And Weaknesses:**

**Strengths**
1. The problem studied, classifying galaxy morphologies, in this paper is quite interesting.
2. The proposed method, which combines a hybrid transformer-convolutional architecture and data augmentation techniques, seems promising effectiveness in tackling this classification challenge.


**Weaknesses**

The main paper is not self-contained, as it lacks experiments. Since this is a tiny paper, I would judge it based on the two-page main content only, ensuring fairness to other "real" tiny papers. This is why I would like to reject this paper.


**Suggested Changes:**

1. Include a more comprehensive set of experiments in the main paper to better support the claims and conclusions.
2. Ensure that the main content of the paper is self-contained.

---

### Comment · Area_Chair_2P7q · 2023-06-06
**Archivial**

 The paper does not include the URM statement

---

### Meta-Review · Area_Chair_2P7q · 2023-04-03

**Recommendation:** Invite to archive
**Confidence:** 3

**Metareview:**

From the reviews, I see the following considerations:
- Clarity: The paper meets the clarity requirements, while a further discussion on the application context would be appreciated
- Correctness: The paper seems correct, but reviewers express concerns about the lack of self-containment since the experiments are reported in Appendix
- Reproducibility: The paper provides enough details to suggest that the work is reproducible and promises the release of the code
- Basic Requirements: The paper follows the basic requirements of the submissions.

The paper collects a positive general opinion while concerns about paper self-containment exist.

The paper reaches the bar of the CCR parameters, but the structure is not optimal, and the Appendix is used to convey some substantial contribution which should be in the main 2-pages manuscript. This weakness is not a sufficient ground for rejection, but addressing it requires a significant paper restructuring, with potential harm to clarity.

**Summary:**

The paper proposes a hybrid transformer-convolutional architecture to predict the Galaxies morphologies. Positive aspects are: obtained SoTA results, addressing an interesting problem clearly presented. The main negative aspects are the lack of self-containment of the paper, some missing discussion on applicability and general context.

**Comments And Feedback To The Authors:**

I would remark on the several positive aspects of the work and encourage the author to continue their work in this direction. Reviews contain several suggestions to improve the presentation, which could help the authors restructure it. In particular, considering changing the title, incorporating experiments in the main draft, and discussing applicative context and challenges seem good recommendations.



**Reason For Not Giving A Higher Recommendation:**

The paper requires a revision to incorporate some missing discussions of applicative context and, in particular, to make the main paper more self-contained. This seems a significant change which might require a substantial rewrite and reorganization of the paper.

**Reason For Not Giving A Lower Recommendation:**

The paper collected positive feedback and demonstrated promising results on an interesting problem. It seems overall clear and reproducible, promising to release the code.

---

### Decision · Program_Chairs · 2023-04-09

Invite to archive